# Densification: A Route towards Enhanced Thermal Conductivity of Epoxy Composites

**DOI:** 10.3390/polym13020286

**Published:** 2021-01-17

**Authors:** Sasan Moradi, Frida Román, Yolanda Calventus, John M. Hutchinson

**Affiliations:** Departament de Màquines i Motors Tèrmics, ESEIAAT, Universitat Politècnica de Catalunya, C/Colom 11, 08222 Terrassa, Spain; sasan.moradi@upc.edu (S.M.); frida.roman@upc.edu (F.R.); yolanda.calventus@upc.edu (Y.C.)

**Keywords:** thermal conductivity, epoxy composites, boron nitride, densification, glass transition, differential scanning calorimetry (DSC)

## Abstract

When an amorphous polymer is cooled under pressure from above its glass transition temperature to room temperature, and then the pressure is released, this results in a densified state of the glass. This procedure applied to an epoxy composite system filled with boron nitride (BN) particles has been shown to increase the density of the composite, reduce its enthalpy, and, most importantly, significantly enhance its thermal conductivity. An epoxy-BN composite with 58 wt% BN platelets of average size 30 µm has been densified by curing under pressures of up to 2.0 MPa and then cooling the cured sample to room temperature before releasing the pressure. It is found that the thermal conductivity is increased from approximately 3 W/mK for a sample cured at ambient pressure to approximately 7 W/mK; in parallel, the density increases from 1.55 to 1.72 ± 0.01 g/cm^3^. This densification process is much more effective in enhancing the thermal conductivity than is either simply applying pressure to consolidate the epoxy composite mixture before curing or applying pressure during cure but then removing the pressure before cooling to room temperature; this last procedure results in a thermal conductivity of approximately 5 W/mK. Furthermore, it has been shown that the densification and corresponding effect on the thermal conductivity is reversible; it can be removed by heating above the glass transition temperature and then cooling without pressure and can be reinstated by again heating above the glass transition temperature and then cooling under pressure. This implies that a densified state and an enhanced thermal conductivity can be induced even in a composite prepared without the use of pressure.

## 1. Introduction

The enhancement of the thermal conductivity of electrically insulating materials used for heat management in many modern electrical and electronic devices has been the objective of a large research effort in recent years. The use of higher frequencies and more compact structures leads to ever increasing service temperatures in such devices, and in order to maintain a satisfactory performance, including a high degree of stability and an acceptable lifetime of operation, it is essential to remove and dissipate the heat generated, usually by conduction through the dielectric layer to a metallic substrate. The dielectric layer must satisfy a number of practical requirements, including good adhesion, ease of processing and low cost, in addition to the physical attributes of electrical insulation and high thermal conductivity. The importance of achieving these objectives can be gauged from the large number of publications devoted to this topic, which have conveniently been collected in several recent comprehensive reviews [1,2,3,4,5,6,7,8].

The requirement for good adhesion leads to epoxy resins being adopted widely as the matrix material, and the present paper is restricted to epoxy composites filled with a suitable inorganic filler. It is well known that the thermal conductivity increases with filler content, but there are numerous other parameters which can influence the thermal conductivity: filler size, shape and type, and surface functionalization to improve the matrix-filler interface are some of the most widely studied aspects, while three-dimensional structuring and other orientation techniques give significant increases in thermal conductivity in preferred directions, though usually at the expense of fabrication complexity. On the other hand, one approach that has not received much attention to date is the application of pressure during cure of the epoxy matrix. In fact, the rather few studies which make use of pressure for the purpose of enhancing the thermal conductivity appear somewhat unsystematic, so that it is difficult to identify what the advantages are of the use of pressure. For example, some workers simply apply pressure to the uncured mixture to improve its compactness before curing at ambient pressure [9,10,11], others only partially cure the sample under pressure followed by a post-cure at higher temperature and only at ambient pressure [12,13], while there are several reports of composites cured under pressure but without the magnitude of the pressure being specified [14,15,16,17,18].

In addition, while there are several reports of samples being fully cured under pressure [19,20,21,22,23,24,25,26,27,28,29], no direct comparison is made with a sample prepared without pressure, and in some cases the preparation method is designed specifically to introduce orientation of the filler particles [25,28,29]. More importantly in the present context, though, these studies do not indicate whether or not the samples were cooled from the cure temperature whilst maintaining the applied pressure. This aspect is important because it determines whether or not the final composite is in a densified state. The purpose of the present paper is to demonstrate the efficacy of densification, and for this it is appropriate to explain the process by which a densified state is achieved.

When a glassy polymer such as an epoxy, or indeed any glass-forming system, is cooled at ambient pressure from above to below its glass transition temperature, the resulting glassy state is one of non-equilibrium. This is illustrated schematically in the volume-temperature (*V*-*T*) or enthalpy-temperature (*H*-*T*) diagram in Figure 1: the initial state is equilibrium at A, the final state is the glassy state B at a temperature which is taken here to be room temperature (*RT*), and the transition at C from the equilibrium liquid-like region (AC) to the glassy region (CB) defines the glass transition temperature, *T*_g_, for the cooling rate used. If the sample is allowed to remain at *RT*, it will display physical aging [30], whereby both the volume and the enthalpy decrease and approach the equilibrium state at that temperature, indicated in Figure 1 by the dashed line extrapolation of the equilibrium liquid-like region. As is well known, this physical aging resulting from the structural (volume, enthalpy) changes in the glass is manifest as a change in many mechanical and physical properties of the material.

If an epoxy composite system is cured at ambient pressure and at a temperature corresponding to point A in Figure 1, which is above the glass transition temperature of the fully cured system, *T*_g__∞_, then on cooling to *RT* it will transform to a glass and reach state B. If, on the other hand, the epoxy composite system is cured under pressure and at the same temperature as before, its initial state after cure will be at D in Figure 1, which we assume here to be above *T*_g__∞_ for the cured system under pressure, state F in Figure 1; note that the increase in *T*_g_ with pressure is much less than 1 °C per MPa [31], and often almost an order of magnitude smaller, which means that the difference between the glass transition temperatures represented by C and F in Figure 1 will be very small under most circumstances.

Once the epoxy composite system has been fully cured under pressure, if it is cooled while maintaining the pressure then it will transform into a glass at point F in Figure 1 and reach state E at *RT*. If the pressure is now released, the sample will recover to state G, the important aspect being that this state is at a volume (and enthalpy) lower than that at state B for the same epoxy system cooled from above *T*_g_ at ambient pressure. In other words, the sample is “densified”, and this has important consequences for many properties, and in particular for the thermal conductivity, as will be demonstrated in this paper.

Several years ago, Senapati et al. [32] investigated the effect of pressure on one such property, the fast ionic conductivity of some silver iodomolybdate glasses, and reported that the ionic conductivity increased with the application of pressure. The application of pressure during use of these glasses in order to profit from the enhanced ionic conductivity would not be feasible in practice, but densification would offer a practical alternative: the densified glasses would retain the enhanced ionic conductivity induced by the high pressure even after the removal of this pressure. Unfortunately, it was later demonstrated [33] that densifying the silver iodomolybdate glasses in fact resulted in lower ionic conductivities, as a consequence of the ionic conductivity actually decreasing on the application of pressure. In fact, this seems more reasonable; the effect of pressure would be to inhibit the pathways for ion transport.

Nevertheless, the procedure of densification remains valid; indeed, one might anticipate the opposite effect to that observed for ionic conductivity when thermal conductivity is considered. In the epoxy composites considered here, the heat transfer occurs preferentially between the highly conducting filler particles, and the effect of pressure might be envisaged as two-fold: both the reduction of the inter-particle distance and the improvement of the particle-matrix interface. The present work investigates just this effect of densification and demonstrates that it presents a highly effective means of enhancing the thermal conductivity of epoxy composites.

## 2. Materials and Methods

### 2.1. Materials

The epoxy resin was a commercial diglycidyl ether of bisphenol-A, DGEBA (Araldite GY240, Huntsman Advanced Materials, Salt Lake City, UT, USA), with a nominal molecular weight per epoxy equivalent (eq) of 182 g/eq, a viscosity of 7000 to 9000 mPa.s at 25 °C, and a density of 1.17 g/cm^3^. A thiol, pentaerythritol tetrakis (3-mercaptopropionate) (Sigma-Aldrich, Saint Louis, MO, USA), with a molecular weight of 488.66 g/mol, a viscosity of 500 mPa.s at 23 °C, and a density of 1.28 g/cm^3^, was used as the cross-linking agent. The cross-linking reaction of the epoxy with the thiol was initiated by a latent initiator, encapsulated imidazole (LC-80, Technicure, A&C Catalysts, Linden, NJ, USA), in the form of powder.

The filler was boron nitride in the form of platelet particles (Saint-Gobain Boron Nitride, Amherst, NY, USA), with a mean size of 30 µm (code PCTP30). According to the manufacturer’s literature [34], these particles have a maximum size of 100 µm, a tap density of 0.6 g/cm^3^, and a specific surface area of 1 m^2^/g. The filler particles were used as received, without any surface treatment.

### 2.2. Methods

#### 2.2.1. Sample Preparation

The epoxy and thiol were mixed by hand in a stoichiometric proportion (approximately 60:40 by weight), and the latent initiator was added in the proportion of 2 parts per hundred resin. While a wide range of filler contents has been used in earlier work on these epoxy composite systems [35,36], for the investigation of the effect of densification in this work only a single filler content was used, namely 70% by weight of boron nitride (BN) with respect to the combined weight of epoxy and BN. The weight percentages of each component in the mixture for this sample, denoted ETLBN30-70, are: epoxy, 24.9%; thiol, 16.6%; BN, 58.0%; LC-80, 0.5%. Making an approximate calculation based on the densities of the constituent components, this corresponds to 44.7 volume% of BN.

In order to obtain compressed samples, the required amount of epoxy-thiol-BN mixture was introduced into a Teflon cylinder of internal diameter 15 mm, outside diameter 60 mm, and height 52 mm. A spring, with a constant of 5.95 kN/m acting on a piston, was used to compress the sample by a measured distance and then locked in place, thus maintaining a constant force. The force on the piston was calibrated by measuring the distance by which the spring was compressed; the maximum pressure that could be achieved in this way was 3.0 MPa. The whole assembly was placed in an air-circulating oven at 70 °C for 3 h to effect the cure under the applied pressure. After cure, the pressure was maintained during cooling, and the cured sample was removed at *RT*. The cured samples, in the form of solid cylinders 15 mm diameter and between 25 and 35 mm in length, were cut in half using a diamond wafering saw to give two smooth and flat surfaces for the measurement of the thermal conductivity.

#### 2.2.2. Thermal Conductivity

The thermal conductivity was measured using the Transient Hot Bridge method (Linseis THB-100, Selb, Germany). A heat pulse is applied to a sensor placed between two surfaces of the sample material, and the thermal conductivity is determined from the temperature change Δ*T* as a function of time [37]. The instrument was calibrated with 5 different standards covering the range from 0.2 to 10 W/mK.

#### 2.2.3. Differential Scanning Calorimetry (DSC)

The DSC instrument (Mettler-Toledo DCS821e, Greifensee, Switzerland) was equipped with a robot sample handler and intracooler (Haake EK90/MT, Vreden, Germany), and was calibrated for both heat flow and temperature using indium. For all experiments, a flow of dry nitrogen at 50 mL/min was used, and the data analysis was made using the STARe software of the instrument.

Powder samples were obtained from the cured composite cylinders by chipping a small amount from the centre of the faces opposite to those used for the measurement of thermal conductivity. Heating scans were made by inserting the encapsulated and weighed samples into the DSC at *RT*, cooling at −20 K/min to 0 °C and then scanning at 10 K/min to 100 °C, well above the *T*_g_ of these composites, which is around 52 °C. A second scan, also at 10 K/min, was performed immediately after cooling to 0 °C at −20 K/min when the first scan had been completed.

#### 2.2.4. Density Measurement

The density of the cured samples was measured by Archimedes method. The samples were first weighed in air at room temperature, and then when immersed fully in ethanol, suspended by a fine thread.

## 3. Results and Discussion

### 3.1. Effect of Pressure on Thermal Conductivity

The thermal conductivities of samples of composition ETLBN30-70 cured under pressures of 1.4 and 2.0 MPa, and then cooled to *RT* whilst maintaining the pressure, are given in Table 1, and the effect of pressure on the thermal conductivity is shown in Figure 2. Two samples were prepared at the pressure of 2.0 MPa to check the reproducibility of the results. In addition, data for samples cured at ambient pressure and at 175 kPa, taken from work published elsewhere [38], are also included for comparison. For the pressure of 175 kPa, the pressure was obtained simply by placing a weight on the piston rather than using the compression spring.

It is evident that the thermal conductivity increases with pressure applied during cure. In fact, the increase in thermal conductivity for the sample cured at 2.0 MPa is remarkable. This increase in thermal conductivity is accompanied by an increase in the density, as can be seen from the values listed in Table 1. With reference to Figure 1, the final states at *RT* of all the samples (S3, S4, S5) cured under high pressure, state G, are confirmed to have higher densities than that of sample S1 cured at ambient pressure, state B. We should point out, however, that although there is some correlation between density and thermal conductivity [38], the latter cannot simply be related directly to the former.

It is interesting to speculate about the mechanism for the enhancement of the thermal conductivity by densification. Figure 3 shows Scanning Electron Microscopy micrographs of the fracture surfaces of two samples: sample S1 cured at ambient pressure, and sample S4 which was densified at a pressure of 2.0 MPa. There can be seen to be a certain amount of consolidation of the composite in the densified state, with a closer connection between matrix and filler particles, implying an improved interface. The closer approach of the individual platelets in the densified sample will also contribute to the enhanced thermal conductivity.

### 3.2. Effect of Densification on Enthalpy

In a similar way, a sample in state G would be expected to have a lower enthalpy, which should be manifest in a DSC scan. The DSC scan for sample S5, cured under 2.0 MPa pressure, is shown in Figure 4 together with the second scan. It can be seen that there is indeed an enthalpy difference between the two scans; the greater area under the first scan, by 0.50 J/g with respect to the second scan, represents the enthalpy difference between states B and G in Figure 1, the lower enthalpy of state G being recovered on heating to above the glass transition region.

A further demonstration of the effect of densification can be seen in the aging behavior of the densified sample S4. This sample, after curing and then cooling to *RT* under the pressure of 2.0 MPa, and then releasing the pressure, was left for two weeks at *RT* before scanning in the DSC. This meant that the densified sample had aged for two weeks at a temperature about 25 °C below its *T*_g_, and the effect of this aging will be manifest as an enthalpy recovery peak in a DSC scan. Such a scan, together with a second scan immediately afterwards, is shown in Figure 5, where it can be seen that there is an area difference between these two scans, the magnitude being 1.23 J/g, with the endothermic peak of the first scan occurring at 52.5 °C. This area difference corresponds to the enthalpy recovered on heating as a consequence of the combined effects of both densification and aging. The same sample S4, after the recovery of both the densification and aging effects, was again subjected to an aging period of two weeks at *RT*, in order to compare the effects of aging of the densified and “undensified” samples. The DSC scan for this aged “undensified” sample is also shown in Figure 5, together with another second scan, which superposes almost exactly on the earlier second scan.

It can be seen that the enthalpy recovery of the aged densified and “undensified” samples is quite different: the latter exhibits a much sharper peak, with a maximum at 55.1 °C, significantly higher than that of the densified sample, and the area difference between first and second scans for the “undensified” sample represents an enthalpy recovery of 1.19 J/g. Given that a part of the enthalpy recovered in the densified and aged sample is that corresponding to the densification, which is 0.50 J/g according to the result for sample S5, the effect of aging in the densified sample is evidently much less than that in the “undensified” sample. The reason for this can be understood by reference to Figure 1. The densified sample, immediately after cure and release of the pressure at *RT*, is in state G, at a lower enthalpy than that of the “undensified” sample in state B. Since state G is closer to equilibrium at *RT*, indicated by the dashed line in Figure 1, the rate of aging of the densified sample will be slower than that of the “undensified” sample.

### 3.3. Reversibility of Densification

An important aspect of the densification is that it is reversible. Consider the densified sample in state G (Figure 1). If it is heated to above its glass transition temperature, and then cooled again to *RT*, it will arrive at state B. This has been demonstrated above by DSC, whereby the second scans are the same, indicating that they always begin from the same state B. The same effect can be seen in the thermal conductivity. The densified sample S5, which had a thermal conductivity of 6.86 W/mK, was reheated to above its glass transition temperature and then cooled again to *RT*, and the thermal conductivity was again measured. It was found now to be 5.44 W/mK, much lower than that of the densified sample, though still larger than that of the “undensified” sample, S1, which was 3.44 W/mK. This implies that the effect of curing under pressure, which results in a more compact epoxy matrix and hence a better matrix-particle interface, but much of which would be recovered on removal of the pressure at the cure temperature, is just one part of the enhancement; the other part results from the “permanent” densification resulting from cooling to *RT* while maintaining the applied pressure. In the present case, for example, the increase in thermal conductivity from 3.44 to 5.44 W/mK would be attributed to the effect of curing under pressure, while the further increase from 5.44 to 6.86 W/mK would be a consequence of the “permanent” densification.

The term “permanent” is written in this way, in inverted commas, because the densification is only “permanent” until the sample is heated above its glass transition temperature, whereupon the densification effect is removed, while the effect on the thermal conductivity of curing under pressure still remains. Likewise, a densified state can be induced in a sample already cured without the application of pressure. With reference to Figure 1, a sample cured without pressure would be in state B when at *RT*. If pressure is applied and the sample is heated above *T*_g_, it will reach a state between A and D; it will not reach state D because there will be no effect of curing under pressure. Subsequent cooling to *RT* under pressure will lead to a state between G and E, and releasing the pressure will result in a final state between B and G, for which one would anticipate a thermal conductivity between that of the sample cured at ambient pressure and that cured under pressure.

This can be illustrated by the results obtained for sample S5, which had previously been densified (6.86 W/mK) and then for which the densification had been removed by heating above the glass transition region (5.44 W/mK). This sample was then returned to the compression cell, a pressure of 2.0 MPa was applied, and the sample was heated above the glass transition region, allowed to equilibrate for several minutes, and then cooled back to *RT* while maintaining the pressure. After releasing the pressure, the thermal conductivity was then measured and found to be 6.76 W/mK. This demonstrates that the densification is reversible and, importantly, that an enhancement of the thermal conductivity can be achieved, retroactively, in a sample already cured without pressure.

### 3.4. Comparison with Literature Values

In Figure 2, the effect of pressure and densification on the thermal conductivity of epoxy-BN composites was demonstrated for a single composition, denoted as ETLBN30-70. The effect of pressure on the thermal conductivity of various epoxy-BN composites has been reported on a number of occasions, but the concept of densification has never been considered in this respect. Furthermore, the use of pressure can be made in many different ways, and in many cases it is not possible to identify clearly the experimental procedure. In order to compare our results presented here, for a single composite composition, with other literature values for a wide range of composite compositions, it is necessary to consider also the effect of BN content on the thermal conductivity. A convenient way in which to do this is to make use of the trend curves in a plot of thermal conductivity as a function of BN content, taken from reference 6 and included in Figure 6.

In the compilation of this figure, the values of the maximum thermal conductivity, together with the corresponding BN content, were taken from more than one hundred references for epoxy-BN composites [6]. In order to simplify and display the overall tendency for the dependence of thermal conductivity on BN content, three trend curves were drawn: an upper trend curve, below which more than 95% of the values fell, including values for both isotropic and anisotropic samples; an intermediate trend curve, which represented an approximation to the upper limit of the isotropic thermal conductivities, thus excluding all those samples for which orientation had been deliberately introduced; and a lower trend curve, below which fewer than 5% of all the values of thermal conductivity fell. These trend curves permit the comparison of our present results with the majority of results reported in the literature.

In the simplest case, pressure can be applied to the uncured mixture at RT simply to consolidate it, to remove voids, or to induce some orientation of the BN particles, the pressure being removed before heating the sample in order to cure it [9,10,11]. On the other hand, there are several reports of samples prepared by curing under controlled pressure-temperature schedules [13,19,20,21,22,23,24,25,26,27,28,29], though some authors do not specify the magnitude of the pressure applied [14,15,16,17,18]. Unfortunately, in no case is there any specific indication of whether or not the cured samples were then cooled under pressure, and hence whether or not they are densified. In contrast, in some cases it is possible to infer that there is no densification, because the samples are post-cured without the application of pressure [12,13,14,16,25]. The maximum values of thermal conductivity taken from these references are plotted as a function of the wt% BN in Figure 6, where they are compared with the upper, intermediate, and lower trend curves discussed above.

There are several results that are worthy of some comment. First, there is only one value, that of He et al. [14], which falls above the upper trend curve. With only 10 wt% of functionalized BN nanoparticles, these authors achieve a thermal conductivity of 1.6 W/mK, in comparison with a value of only 0.5 W/mK when the BN particles are not surface treated. This result is particularly remarkable as pressure (though an unspecified value) is applied during cure at 100 °C, and then the sample is post-cured without pressure at 150 °C, implying that the final composite is not densified.

Second, many of the values which fall in the region between the intermediate and upper trend curves are a consequence of orientation, the values included in Figure 6 being those in the preferentially oriented direction [9,13,25,28]. Of the remaining values lying in this same region, the results of Jang et al. [15], the present work, and Xu and Chung [19] all follow approximately the same dependence of thermal conductivity on BN content. In the work of Jang et al. [15] and Xu and Chung [19], the BN particles were surface treated, which often leads to improved thermal conductivity, whereas in the present work, the particles were untreated. It is possible, therefore, that the disadvantage of using untreated particles in the present work is compensated by the effects of densification.

The data presented in Figure 6 are for epoxy-BN composites, but for comparison we have also included the data of Tang et al. [27] and Zhang et al. [29], for both of which the composite was a hybrid in which the epoxy matrix was filled with both glass fibres (or cloth) and BN particles. The thermal conductivity enhancement of the epoxy matrix is therefore due to both the glass and the BN, while the manufacturing procedure introduced significant orientation. Tang et al. [27] report that the addition of 20 wt% BN increases the thermal conductivity of the epoxy/glass/BN hybrid by factors of 5.4 and 3.0 in the in-plane and through-plane directions, respectively. To compare this result with our own values, we have applied an average factor of 4.2 to the unfilled epoxy (0.23 W/mK) to obtain the value of 0.97 W/mK plotted in Figure 6, which falls just on the intermediate trend curve. Likewise, Zhang et al. [29] report an increase in the thermal conductivity by a factor of 2.5, in both the in-plane and through-plane directions, for epoxy/glass/BN hybrids filled with 15 wt% BN. Applying the same factor to the unfilled epoxy (0.2 W/mK) gives a value of 0.50 W/mK for the thermal conductivity of an epoxy/BN composite for comparison with our own results. This is plotted in Figure 6, where it can be seen to fall close to the intermediate trend curve. The application of pressure during cure, 5 MPa in both these cases [27,29], appears therefore to raise the thermal conductivity to the level of the intermediate trend curve, but these values still fall below that which can be obtained by densification.

Finally, the very high value of 10.5 W/mK at 58 wt% BN reported by Moradi et al. [38] corresponds to BN in the form of agglomerates, densified at 2.0 MPa pressure. The effect of pressure on agglomerates has been suggested to be different from that on platelets; whereas pressure brings the matrix and platelets closer and hence improves the interface, the agglomerates are deformed by the pressure such that the surface area of contact between matrix and filler, and between one filler particle and another, is considerably increased, with a resulting dramatic increase in the thermal conductivity [38].

## 4. Conclusions

It has been demonstrated that the application of pressure during the cure of composites of epoxy and boron nitride has two separate effects: curing under pressure results in a more compact matrix and a better matrix-filler interface, whereas subsequently cooling from the cure temperature to room temperature while maintaining the pressure results in densification. Both effects contribute to an enhancement of the thermal conductivity of the composite. The densification is reversible: heating the densified sample, at ambient pressure, to above its glass transition temperature removes the densification, and the thermal conductivity is consequently reduced. Likewise, though, the densified state can be reintroduced by heating the sample again to above its glass transition temperature, applying pressure, and then cooling to room temperature before releasing the pressure, whereupon the thermal conductivity returns to its original densified value. Comparison is made between these results and others reported in the literature for samples prepared under pressure.

## Figures and Tables

**Figure 1 polymers-13-00286-f001:**
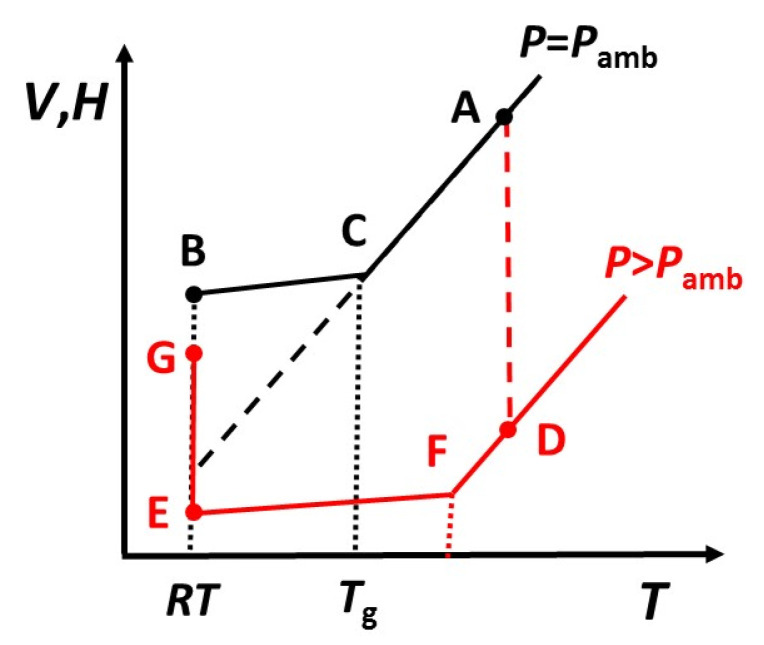
Schematic illustration of the glass transition region and the effect of pressure.

**Figure 2 polymers-13-00286-f002:**
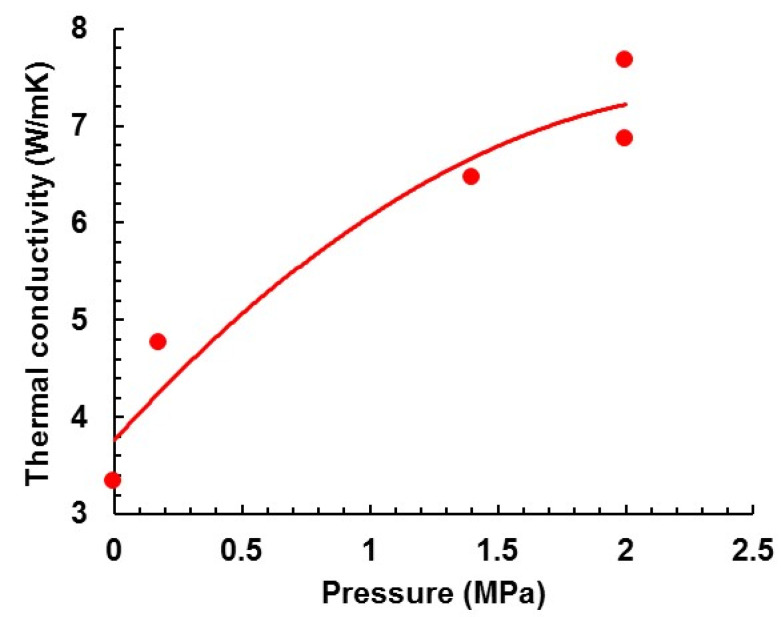
Dependence of thermal conductivity on pressure.

**Figure 3 polymers-13-00286-f003:**
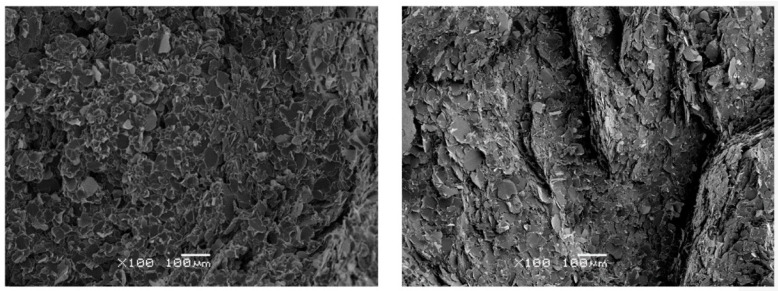
Scanning Electron Microscopy micrographs of two samples of composition ETLBN30-70: left-hand photo for sample cured at ambient pressure, right-hand photo for sample densified at 2.0 MPa. Magnification ×100, scale bar 100 µm.

**Figure 4 polymers-13-00286-f004:**
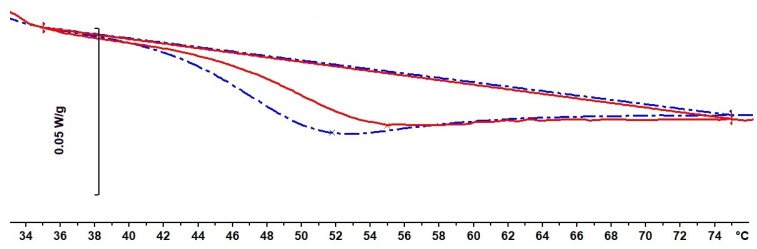
Differential Scanning Calorimetry (DSC) scan at 10 K/min of sample S5 (dash-dotted blue curve), densified at 2.0 MPa, followed by second scan (full red curve). Baselines for the determination of the areas are constructed from 35 °C to 75 °C. Endothermic direction is downwards.

**Figure 5 polymers-13-00286-f005:**
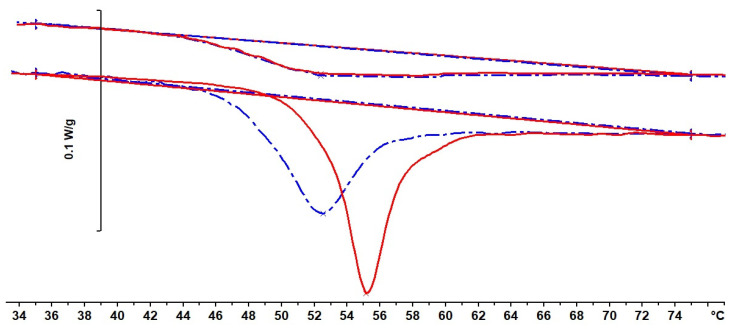
Comparison of DSC scans after aging for two weeks at *RT* for densified (dash-dotted blue curve) and “undensified” (continuous red curve) samples, together with their respective second scans. Baselines for the determination of the areas are constructed from 35 to 75 °C, and the second scans have been displaced vertically for clarity. Endothermic direction is downwards.

**Figure 6 polymers-13-00286-f006:**
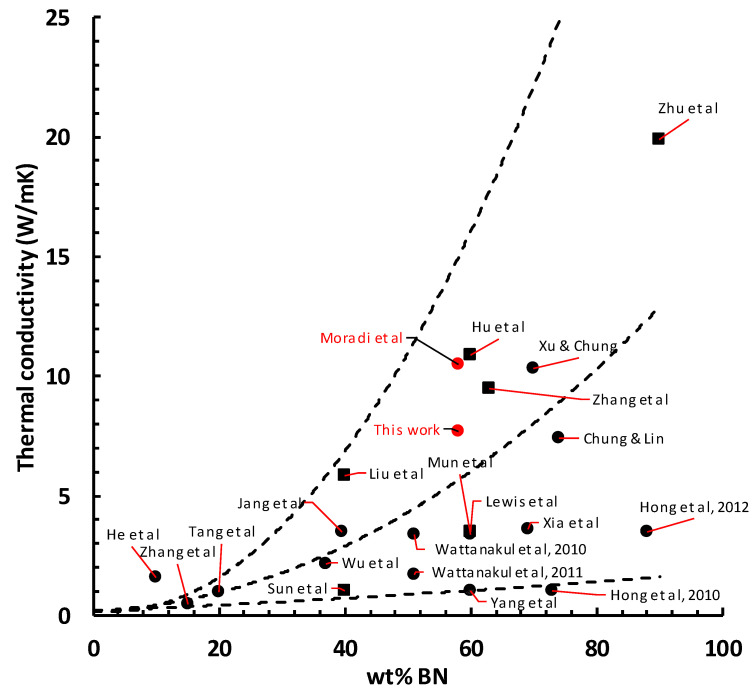
Thermal conductivity as a function of boron nitride (BN) content (wt%) for epoxy-BN samples prepared under pressure. Dashed lines represent upper, intermediate, and lower trend curves [6]. Squares represent oriented samples.

**Table 1 polymers-13-00286-t001:** Thermal conductivity and density of ETLBN30-70 samples.

Sample	Pressure (MPa)	Thermal Conductivity (W/mK)	Density (g/cm^3^)
S1	ambient	3.34	1.55
S2	0.175	4.77	1.56
S3	1.4	6.47	1.63
S4	2.0	7.67	1.71
S5	2.0	6.86	1.73

## Data Availability

The data presented in this study are available on request from the corresponding author.

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
