# Peer review of "Densification: A Route towards Enhanced Thermal Conductivity of Epoxy Composites"

_polymers, 2021, doi:10.3390/polym13020286_

Round 1

Reviewer 1 Report

polymers-1055482(Major revision)

In this work, the authors reported an epoxy composite system filled with boron nitride particles has been shown to increase the density of the composite, reduce its enthalpy and, most importantly, significantly enhance its thermal conductivity through the thought of when an amorphous polymer is cooled under pressure from above its glass transition temperature to room temperature, and then the pressure is released, this results in a densified state of the glass. The authors also found the densification and corresponding effect on the thermal conductivity is reversible, it can be removed by heating above the glass transition temperature and then cooling without pressure, and can be reinstated by again heating above the glass transition temperature and then cooling under pressure. The paper can be consideration for publication in this Journal provided that the following issues must be further addressed.

  1. It can be seen from Table 1 that the density and thermal conductivity increases with the increase of pressure. Can the density be considered as the direct factor affecting the thermal conductivity?
  2.  The mechanism of densification improving thermal conductivity of epoxy composites should be supplemented.
  3.  Please further highlight the novelty of the article. In this work, the authors reported BN/epoxy resins composites. To our knowledge, there are many relative reported relative works about thermally epoxy resins composites. The authors are encouraged to read recently published relative papers, especially for epoxy matrix (e.g. J Mater Sci Technol, 2021, 68: 209; Compos Sci Technol, 2020, 200: 108456; Scientific Reports, 2020, 10: 14926; Composites Part A, 2020, 129, 105696) and BN or BNNS fillers (e.g. Composites Part B, 2020, 187:107855; ACS Appl Mater Inter, 2020, 12: 1677; Journal of Polymer Research 2020, 27: 212; Composites Part B, 2020, 185: 107784), etc, to reveal the advantages or novelty of your work, finally to further highlight the theme.
  4. Is the increase in density due to the increase of thermally conductive filler content? If so, the increase of thermal conductivity is only due to the increase of filler content, not the densification.
  5. Figure 1 needs to be annotated with references if it is an existing theoretical result.
  6. Why no density data of samples C and E in Table 1?
  7. In the experimental, the thermal conductivity was measured using the Transient Hot Bridge method (Linseis THB-123 100, Selb, Germany). Why not choosing transient plane-source or Hot Disk method described in the following review paper (Compos Commun, 2019, 16: 5-10), which may be more suitable to testing the thermal conductivity coefficient.
  8. The References are too old and should be supplemented and replaced. Furthermore, the cited references can hardly reflect the main publications on which the work is based. Some important relative works should be reviewed or compared. 9. There are some grammar and word mistakes in the manuscript. Please go through the manuscript carefully again.

Reviewer 2 Report

The paper entitled "Densification: a route towards enhanced thermal conductivity of epoxy composites" reports about an interesting study dealing with the application of a densification treatment on epoxy-based composites containing boron nitride particles.

In my view, the following concerns have to be solved, bofore recommend the publication of the manuscript:

  • the Abstract of the manuscript is too general; i suggest to the Authors to re-written it, adding some sentences on the experimental procedure followed in the work and on the obtained results;
  • the Introduction part appears very poor; the discussion should be enriched, adding some references about similar investigations already reported in the literature;
  • the results coming from thermal conductivity measurements look very interesting, but a discussion on the obtained data should be added in paragraph 3.1;
  • in general, the manuscript should be enriched with some additional discussion and/or comments about the obtained results and their correlation with the material structure.

Author Response

Please see the atachment.

Round 2

Reviewer 1 Report

The cited references can hardly reflect the main publications on which the work is based. Some important and similar relative works about epoxy thermal conductivity composites (Composites Part A, 2018, 107: 217; Compos Sci Technol, 2020, 187: 107944; Chemical Engineering Journal, 2020, 397: 125447; Polymers, 2019, 11: 1548; Chinese J Polym Sci, 2020, 38: 730;) filled with BN fillers or BNNS fillers should be compared. Therefore, the authors are encouraged to provide the corresponding table containing the thermal conductivity Coefficient to reveal the advantages of your work.

Reviewer 2 Report

I suggest the publication of the manuscript as it stands, as the Authors modified it following the suggestions of the Reviewer.

Author Response

We thank the Reviewer for recommending publication.